# Evolution of multiple cell clones over a 29-year period of a CLL patient

Zhikun Zhao[1,2,3,*], Lynn Goldin[4,*], Shiping Liu[1,5,*], Liang Wu[1,*], Weiyin Zhou[6,*], Hong Lou[6,*], Qichao Yu[1,7], Shirley X. Tsang[8], Miaomiao Jiang[1,3], Fuqiang Li[1], MaryLou McMaster[4], Yang Li[1], Xinxin Lin[1], Zhifeng Wang[1], Liqin Xu[1], Gerald Marti[9], Guibo Li[1,10], Kui Wu[1,10], Meredith Yeager[6], Huanming Yang[1,11], Xun Xu[1,**], Stephen J. Chanock[4,**], Bo Li[1,**], Yong Hou[1,10,**], Neil Caporaso[4,**] & Michael Dean[1,4,**]

Chronic lymphocytic leukaemia (CLL) is a frequent B-cell malignancy, characterized by recurrent somatic chromosome alterations and a low level of point mutations. Here we present single-nucleotide polymorphism microarray analyses of a single CLL patient over 29 years of observation and treatment, and transcriptome and whole-genome sequencing at selected time points. We identify chromosome alterations 13q14 − , 6q − and 12q + in early cell clones, elimination of clonal populations following therapy, and subsequent appearance of a clone containing trisomy 12 and chromosome 10 copy-neutral loss of heterogeneity that marks a major population dominant at death. Serial single-cell RNA sequencing reveals an expression pattern with high FOS, JUN and KLF4 at disease acceleration, which resolves following therapy, but reoccurs following relapse and death. Transcriptome evolution indicates complex changes in expression occur over time. In conclusion, CLL can evolve gradually during indolent phases, and undergo rapid changes following therapy.

[1] BGI-Shenzhen, Shenzhen 518083, China. [2] State Key Laboratory of Bioelectronics, Southeast University, Nanjing 210096, China. [3] School of Biological Science and Medical Engineering, Southeast University, Nanjing 210096, China. [4] Division of Cancer Epidemiology and Genetics, National Cancer Institute (NCI), National Institutes of Health (NIH), Bethesda, Maryland 20892, USA. [5] School of Life Sciences, Sun Yat-sen University, Guangzhou 510006, China. [6] Cancer Genomics Research Laboratory, National Cancer Institute, Division of Cancer Epidemiology and Genetics, Leidos Biomedical Research Inc., Bethesda, Maryland 20892, USA. [7] BGI-Education Center, University of Chinese Academy of Sciences, Shenzhen 518083, China. [8] Biomatrix, Bethesda, Maryland 20849, USA. [9] Center for Devices and Radiological Health, Food and Drug Administration, Silver Spring, Maryland 20993, USA. [10] Department of Biology, University of Copenhagen, Copenhagen 1599, Denmark. [11] James D. Watson Institute of Genome Sciences, Hangzhou 310058, China. * These authors contributed equally to this work. ** These authors jointly supervised this work. Correspondence and requests for materials should be addressed to B.L. (email: libo@genomics.cn) or Y.H. (email: houyong@genomics.cn) or to N.C. (email: caporasn@mail.nih.gov) or to M.D. (email: deanm@mail.nih.gov).

Chronic lymphocytic leukaemia (CLL) is the most common B-cell malignancy in the US, Canada and Western Europe, and remains an incurable disease[1–5]. Recurrent somatic alterations include deletions of chromosomes 11q, 13q14 and 17p, and trisomy 12 (refs 6,7) and point mutations in SF3B1, NOTCH1 and TP53 (refs 8–10). CLL represents an interesting model to study cancer progression, therapy response and relapse, as the disease is often detected many years before the initiation of treatment, and patients survive for a considerable time. We took advantage of an extremely rare situation of having yearly viably frozen tumour cells from a patient over the past 18 years of her 29-year disease course. By performing single-nucleotide polymorphism (SNP) array analysis at 16 yearly time points,
as well as single-cell whole-genome sequence (WGS) and transcriptome, we have a detailed picture of molecular changes over time. During this time period, the patient had a 9-year period of indolent disease, a marked rise in white blood cell (WBC) counts, and multiple years of cytotoxic therapy with a moderate disease progression, followed by more rapid progression and chronic infections, and death. The resulting analysis provides an unparalleled look at cancer evolution over nearly 20 years.

## Results

**Patient description**. A female CLL patient was diagnosed in 1972 at age 47, with no evidence for cytogenetic abnormalities. We divide her disease into an early phase of observation (no treatment of disease lasting until 17 years after diagnosis), a middle phase (moderate disease progression requiring treatment, 18–25 years) and a late phase (disease progression, chronic infections and death, 26–29 years; Fig. 1b; Supplementary Fig. 1). Cytotoxic therapy (chlorambucil, an alkylating agent) administered in year 16, 22/23 resulted in a short-lived remission; eventually the patient progressed and died of her disease 29 years after diagnosis at age 76.

**SNP microarray analysis**. To assess the sequence of changes in chromosomal abnormalities, we performed microarray analysis on tumour cells at 16 time points over 21 years (Figs 1a and 2; Supplementary Figs 2–6). There were no detectable aberrations at year 8 or 27 that reflect early disease and remission stages, respectively. Chromosome 6q and 13q deletions, copy-neutral loss of heterogeneity (LOH) on 10p and gain on chromosome 12 (years 10–12, 14, 17, 19–26 and 28) with at least two different events at each time point detected (Fig. 2; Supplementary Fig. 7).

Chromosome 13q− was found for all 14 time points with alterations. The focal deletion region 13q14.3 was identified at early time points and persisted to the end (years 10–28) and coincided with 6q− and 12q+ alterations. The large 13q deletion involving the RB1 gene (13q14.2) was found at later time points (years 20–25; Fig. 2; Supplementary Fig. 7), and may reflect disease progression and clonal selection. Chromosome 12 trisomy was found at later time points (years 25–28) and coincides with chromosome 10p copy number LOH (CNLOH; Fig. 2).

**Whole-genome and single-cell sequencing**. To better understand the genomic changes, we sequenced the whole genome of unsorted peripheral blood mononuclear cells (PBMCs) DNA samples from years 10, 14, 21, 23, 24, 26 and 28 (Supplementary Table 1). The copy number variation (CNV) patterns were highly consistent with the results of the SNP microarray (Supplementary Figs 8,9). Interestingly, 6q deletion is always present with the 12q duplication (Fig. 3a). Copy-neutral LOH of chromosome 10p was found for four later time points (years 25–28) accompanied by

whole chromosome 12 duplication (Fig. 3a). The disappearance of 6q deletion and 12q duplication, and the appearance of 10p CNLOH and 12 trisomy may reflect clonal selection in response to treatment. A low number of apparent somatic mutations were detected. These include S219C in MYD88 in 11–30% of reads in years 21, 23, 24 and 26, but undetectable in years 10, 14 and 28; and G49S in MED12 in 11% of reads only in year 26 (Supplementary Table 2). Both of these mutations have been previously detected in CLL[10].

To characterize the chromosome abnormalities at the single-cell level, low-coverage WGS from years 23 and 28 cells was generated (Fig. 1a; Supplementary Fig. 2; Supplementary Table 3)[11,12]. The data reveals clusters of tumour cells with chromosome 13q−, chromosome 6q− and partial or complex trisomy of chromosome 12 (Fig. 3b; Supplementary Fig. 10). The CNVs of 6q−, 12+, trisomy 12, focal 13q− and large 13q− detected by SNP microarray were all present in the single cells. However, single-cell analysis allows the CNVs to be resolved into five distinct sub-populations: normal-like; focal 13q− only; focal 13q− and large 13q−; 6q− and 12q+, 13q14.3−; and trisomy 12 only (Fig. 3b). The chromosome 12q gain at year 23 always co-occurred with the 6q deletion consistent with the results of SNP microarray. These two CNVs were also accompanied by the focal 13q deletion. The remaining cells harboured either the large and focal 13q deletion or only the focal 13q deletion, for a total of three distinct cell populations. In year 28, the chromosome 12 trisomy emerged in the absence of other CNVs, indicating a novel origination of this clone. By combining the CNVs profiles from SNP array and single-cell sequencing, we reconstructed that the focal 13q14.3 deletion appears first, followed by the co-occurring 6q− and 12q+ events (Fig. 3a). A new clone with the large 13q14.2-14.3 deletion is first detected in year 20. Subsequently, the patient had multiple infections and elevated WBC counts, and after cytotoxic therapy (year 16, 22/23) and splenectomy in year 25, all chromosomal abnormalities were undetectable except the focal 13q13.3 deletion. In year 24, both CNLOH of chromosome 10 and trisomy 12 emerge, both of which persist until year 28.

**Single-cell transcriptome analysis**. To investigate gene expression profiles over time, we performed single-cell RNA transcriptome sequencing (RNA-seq)[13] on 300 unsorted tumour cells from the year of year 10, 20, 23, 26 and 28 (Supplementary Fig. 13; Supplementary Table 4). Unsupervised hierarchical clustering and principal component analysis (PCA) yielded six clusters (Fig. 4a; Supplementary Fig. 14). Cells from the earliest time point (year 10) are almost exclusively found in cluster D suggesting that this is the earliest detectable profile, and several cancer-related genes, including MAPK4 (ref. 14), ERBB4 (ref. 15) and PDGFRA[16] are in this cluster (Fig. 4b; Supplementary Tables 5 and 6). In year 20, the cells adopt almost exclusively cluster F that contains several transcription factors involved in stem cell regulation, such as JUN, FOS, KLF4, KLF6 and CDKN1A, the MYD88 cascade (FOS, JUN, NFKBIA and RPS6KA5) or downstream signalling of the B-cell receptor (BCR; REL, CDKN1A and NFKBIA)[17,18] (Fig. 4b; Supplementary Tables 5 and 6).

Beginning in year 23, the cells developed a greater diversity of expression profiles divided into two major branches with clusters A, B and C, predominant in post-treatment samples. The cells from year 23 contain mostly profiles B and C; and in year 26, when the patient was in remission the cells largely reverted to cluster D. Finally, in year 28, at relapse, the cells again developed a great diversity with 56% of the cells adopting expression patterns clusters A and C, and 36% of cells a new expression profile, cluster E (Table 1; Fig. 4b; Supplementary Table 5).

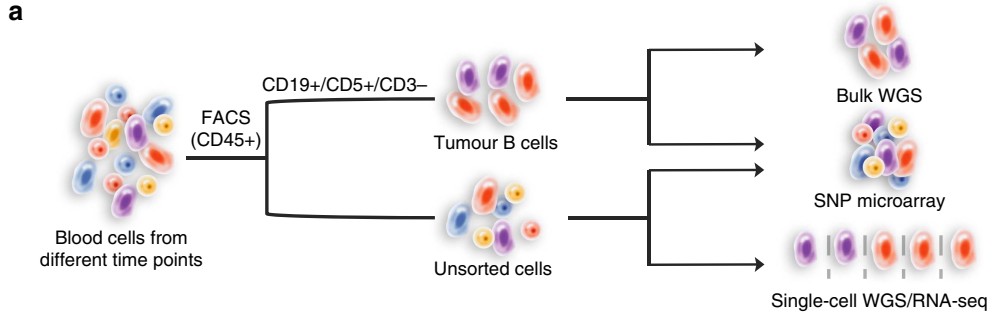

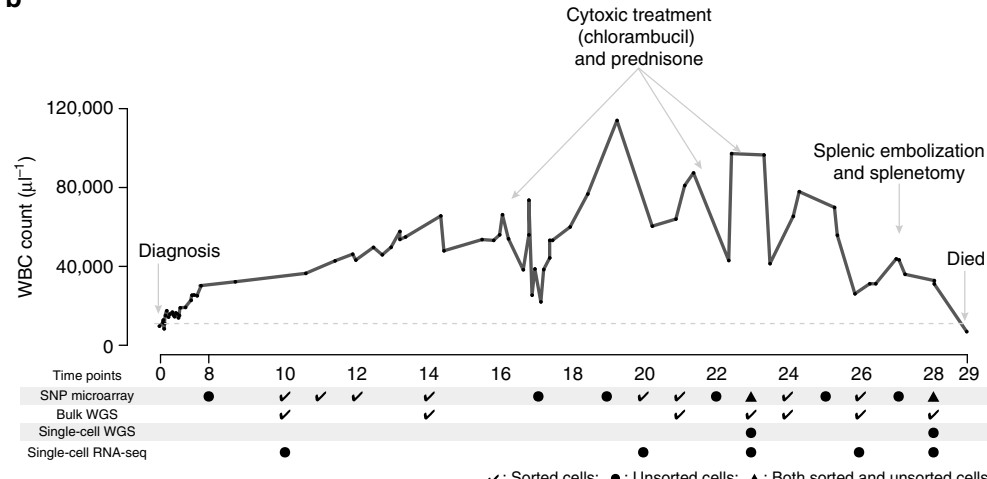

**Figure 1 | Sample and clinical information. (a)** Schema of the sample isolation and the sequencing strategy. FACS, fluorescence-activated cell sorting. WGS, whole-genome sequencing. RNA-seq, RNA sequencing. **(b)** Clinical information and corresponding samples analysed. Top, white blood cell counts (WBC) from diagnosis to year 29 following diagnosis. The dashed grey line indicates the normal upper level of WBC. Additional clinical information is shown in Supplementary Fig. 1. Bottom, the matched sequencing samples.

Therefore, the single-cell expression analysis reveals a diverse evolution of expression profiles over time and during treatment, remission and disease relapse.

To further explore evolutionary models based on the single-cell RNA-seq data, we performed cell fate decision analyses using the Monocle program[19] to reorder the cells by their differential expression genes profiles. This pseudo-temporal ordering analysis decomposed the 300 cell profiles into three trajectories (Fig. 4c,d). The first trajectory is dominated by the oldest cells from year 10. The second trajectory contains mostly cells from year 20 and the third trajectory cells from year 23 followed by year 28 cells. The cells at the end of the third trajectory are mostly the year 28 cells from cluster E.

The program Monocle groups genes with similar expression patterns along the pseudo-temporal trajectories. Several of these groups contain a high–low–high expression pattern, with high expression in early stages of leukaemia development, low expression after therapy and remission, and high upon relapse (Supplementary Fig. 15; Supplementary Table 7). These clusters are enriched in cancer-related pathways, such as BCR signalling, epidermal growth factor receptor (EGFR) signalling and extracellular signal-regulated kinase (ERK) signalling. Another group (group 7) is highly significantly enriched in cell cycle-related cyclin A and E genes, and NF-KappaB genes involved in B-cell development. This group displays low expression during the initial stages of leukaemogenesis and the highest expression at the early portion of trajectory 3.

## Discussion

In summary, we have performed multiple microarray, bulk and single-cell DNA and RNA analyses across the evolutionary lifespan of a single CLL patients' disease, response to therapy, relapse and death. We can divide the disease process into three phases:

In the early phase, at the first time point, year 10 after clinical presentation of CLL, a focal deletion of chromosome 13q14.3 was present along with a deletion of a portion of chromosome 6 and a gain of chromosome 12q. Single-cell analysis showed that the $6q-$ and $12q+$ alterations are present in the same cells, which lack the chromosome 13 deletion (Fig. 5). This is consistent with data that $13q-$ cells can occur many years before CLL diagnosis (manuscript in preparation). The mosaic fraction of the $6q-$ and $12q+$ alterations change in parallel over the period from year 10–19, consistent with the cell population containing these variants varying over time.

In the middle phase, in year 20, the chr13 deletion expanded to include the *RB1* gene, an event associated with poorer prognosis[20], and a mutation in *MYD88* (S219C) is also detected. These events are accompanied by a marked alteration in the gene expression pattern to expression cluster F. Elevated expression of *FOS* and *JUN* transcription factors and oncogenes, *KLF4* (refs 21,22), one of the critical stem cell induction genes, as well as *REL*[23], *CDKN1A*[24] and B-cell oncogenes are observed. This corresponds to the peak of a marked rise in WBC counts with the expansion of the chr13 deletion to include *RB1* may lead to the altered regulation of one or more transcription factors, such as *FOS/JUN* and *KLF4* accelerating the growth of the leukaemic cells.

In the late phase, in year 22, further cytotoxic therapy was administered and in year 23 the expression pattern changed to one of high diversity with multiple different expression clusters

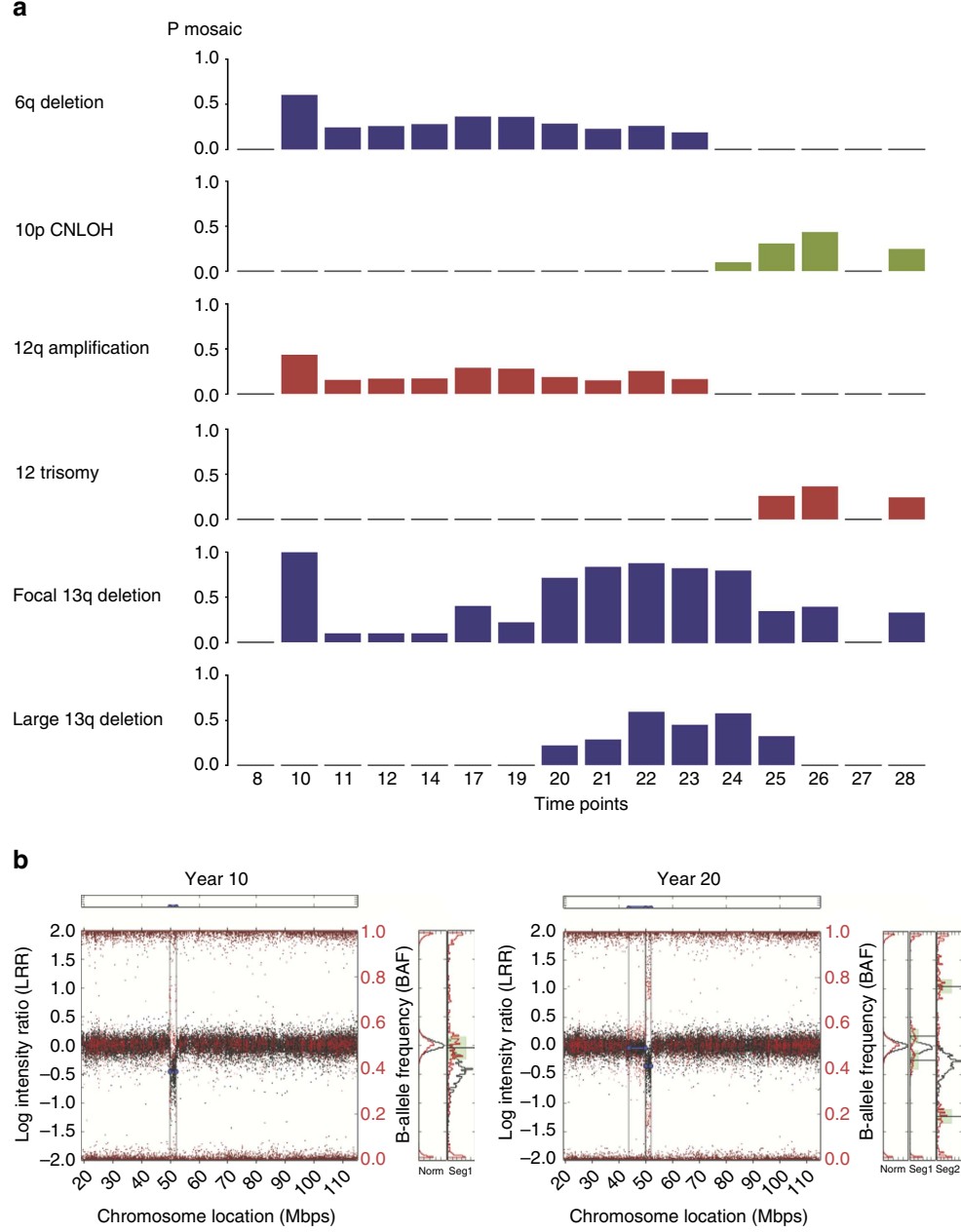

**Figure 2 | Representative CNV profiles detected by SNP microarray.** (**a**) The CNV profiles of 16 time points during the years 8–28 from diagnosis are shown. (**b**) The SNP array plots of 13q − in years 10 and 20 are shown.

present in different cell populations. Furthermore, in year 24 we see the appearance for the first time of a CNLOH event on chr10 and a complete trisomy of chr12, consistent with these events being concurrent in the same population. The single-cell sequencing shows that the trisomy 12 event occurred on an otherwise normal chromosome background.

The patient required a splenectomy year 25, and in year 26 the gene expression pattern reverts to cluster D, the expression pattern present in the indolent phase. In year 27, most of the chromosome abnormalities become undetectable, and the patients' disease was stable. However, this state is short lived, and in year 28 all the chromosome abnormalities present before remission are detectable again. The patient suffered from increasing and recurrent respiratory tract infections and died in year 29. The gene expression pattern in year 28 shows a similar diversity in gene expression clusters as in 23, but the proportion

of cells in each cluster has changed markedly. Therefore, we can follow the progression of the patients' disease at the molecular level along with the clinical changes or remission and relapse.

To further understand the alterations in gene expression over time we used a new method, Monocle[19], to analyse temporal changes in gene expression. Because cells do not synchronously move through expression states, this analysis can be used to reorder the cells based on both the time and expression state identifying genes that are coordinately expressed. This analysis showed cells from trajectory 1 contain many of the cells of year 10 and early time points. One group of cells branches off and stops (trajectory 2), and we interpret this as being cells eliminated by the therapy. Nearly, all the year 26 (cluster D) cells are in here. The third trajectory is composed mostly of cells with complex expression profiles from years 23 and 28. The final group of cells is almost exclusively from the year 28 time point and expression cluster E. Cluster E is a very

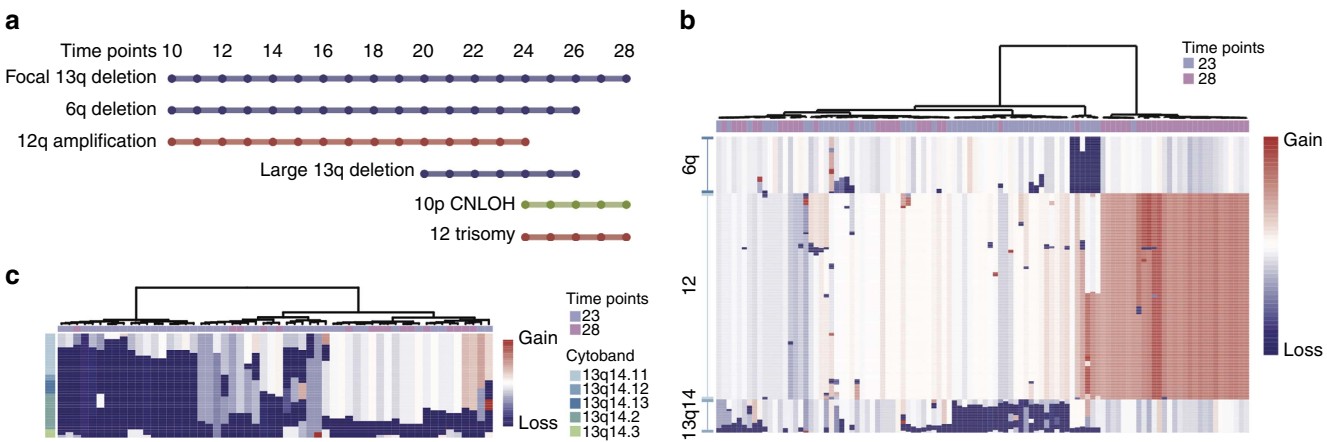

**Figure 3 | CNV profiles detected by WGS and single-cell WGS.** (**a**) The timeline of CNVs detected by combined SNP microarray and WGS. CNLOH, copy-neutral loss of heterogeneity. (**b**) The CNV profiles of 6q, 12 and 13q14 by single-cell WGS. Two cells were removed (Supplementary Figs 11 and 12). (**c**) The CNV profiles of the 13q14 region. The cells with a normal karyotype in this region were removed. The CNV profiles of all cells were showed in Supplementary Fig. 10b.

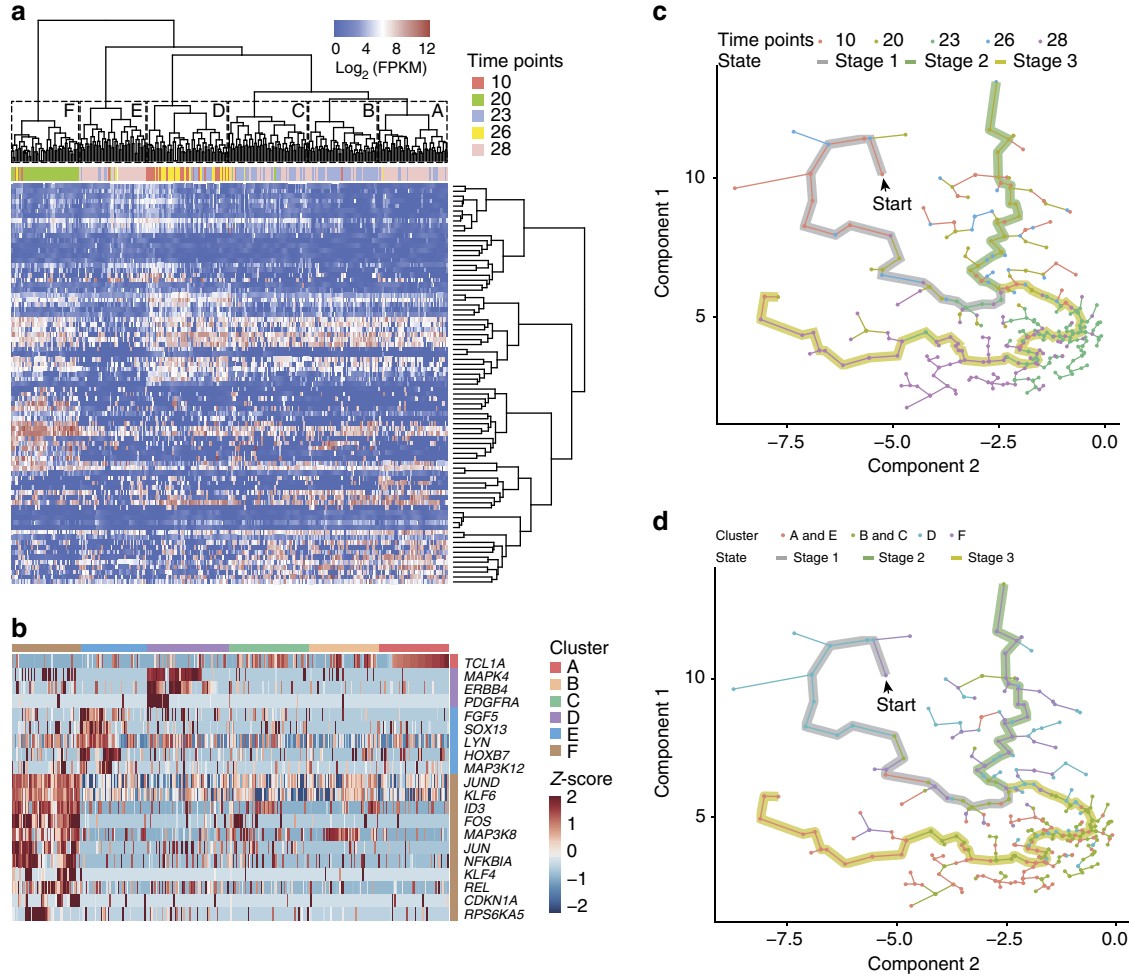

**Figure 4 | Dynamic changes revealed by single-cell RNA-seq analysis.** (**a**) Hierarchical clustering of single-cell RNA-seq data from 300 single cells of five time points. Each column represents a single cell, and each row represents a gene. (**b**) The Z-scores of cells from different time points in different clusters. (**c**,**d**) Cell expression profiles in pseudo-temporal ordering. Points represent single cells. Lines connecting points represent the edges of the minimum spanning tree by Monocle program. The thick lines represent the main path of the pseudo-temporal ordering.

minor component (2–7%) of the cells in year 10–26, but is 36% of the cells in the year 28. This suggests that these cells were rapidly proliferating at the end stage of disease. The cell population that contains the trisomy 12 and most likely the CNLOH event on chromosome 10, are likely the cluster E-expressing cells, containing *FGF5*, *LYN*, *SETD2* and *IL17RD*. *LYN*[25] is a tyrosine kinase that is

**Table 1 | Percentage of single cells in each expression cluster by date.**

| Cluster | Year 10 (%) | Year 20 (%) | Year 23 (%) | Year 26 (%) | Year 28 (%) |
|---------|-------------|-------------|-------------|-------------|-------------|
| A | 0 | 0 | 11 | 3.7 | **36** |
| B | 3.2 | 0 | **41** | 0 | 8.7 |
| C | 6.5 | 4.2 | **30** | 7.4 | 20 |
| D | **77** | 0 | 11 | **82** | 0 |
| E | 3.2 | 2.2 | 6.5 | 3.7 | **36** |
| F | 9.7 | **94** | 0 | 3.7 | 0 |

The percentage of single cells in each expression cluster as determined in Fig. 4a is shown at each time point tested. Bolded values are the highest at that date.

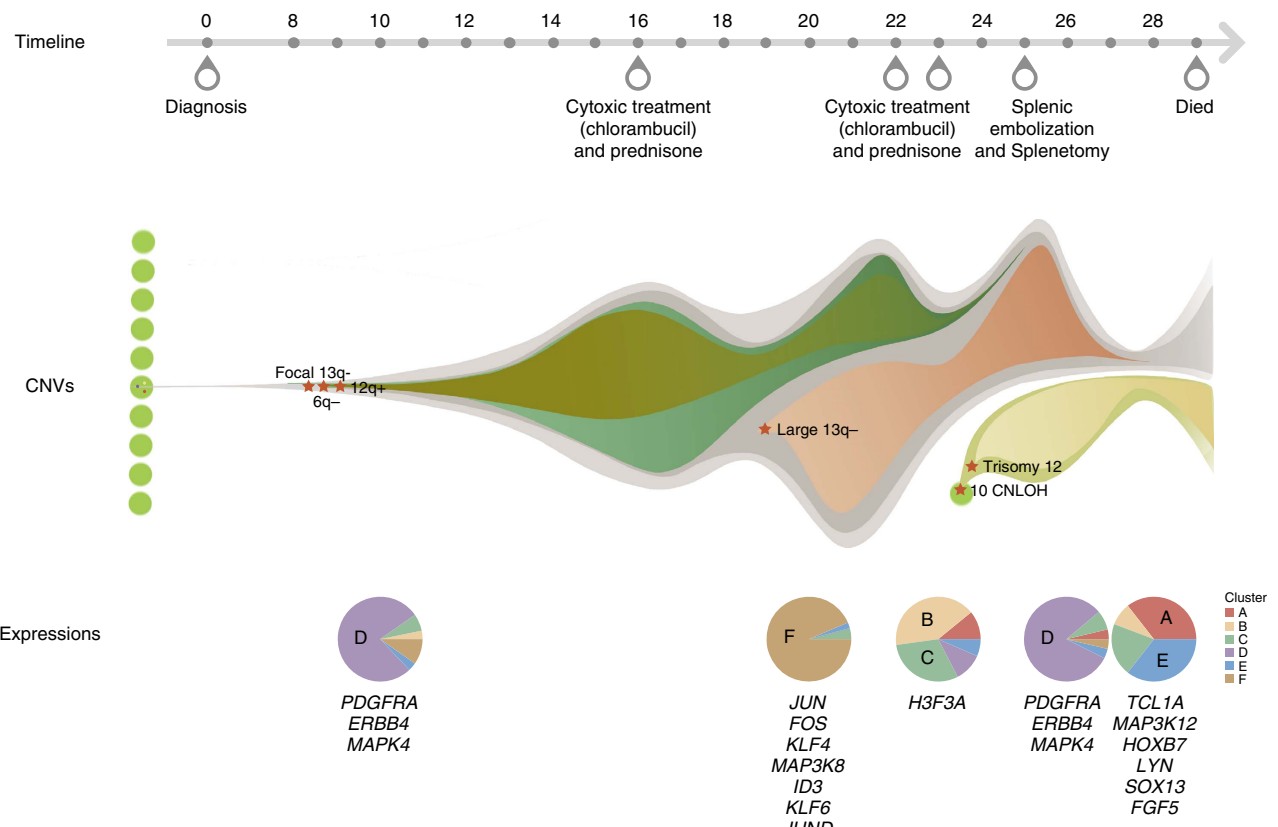

**Figure 5 | The inferred tumour evolution path of the CLL patient.** The upper panel represents the timeline of the main treatments. The middle panel represents the inferred CNVs evolution model and the lower panel represents the percentage of cells from different clusters at expression level.

expressed in B cells and mediates response from the BCR. *LYN* is located on chromosome 12, and the trisomy 12 event may contribute to this overexpression.

Deletions of chromosome 13 are the single most common somatic events in CLL. The focal 13q14.3 deletion contains two long-non-coding RNA genes *DLEU1* and *DLEU2* and is thought to disrupt the expression of *miR-16-1* and *miR-15a* (ref. 26). This deletion is a good prognosis factor for CLL patients. However, large deletions that also contain *RB1* are poor prognostic factors, especially when homozygous. While it has been assumed that the focal deletions expand to become the larger deletions, our data (Fig. 3c) provide direct evidence for this, as we can see individual single cells with transition states from the focal to expanded deletion. Interestingly, homozygous large deletion cells are almost completely absent from the single cells in year 28, indicating that they were eliminated by the therapy.

The study is limited by involving a single patient, and therefore the conclusions to be drawn to other CLL subjects would require

other analyses. However, this patient does have most of the commonly occurring chromosomal aberration found in CLL and therefore is a typical case in that respect. Our array and single-cell analyses over so many time points allows the reconstruction of the distribution and evolution of these chromosome abnormalities during the disease process. This patient had an indolent course of disease for many years, but once the disease required therapy, clinical progression proceeded in a typical manner. The most unique findings are the dramatic changes in gene expression patterns seen in single cells over time. Again it remains to be seen if this is typical of CLL from future single-cell multiyear analyses.

In conclusion, we have performed, to our knowledge, the most detailed and extensive analysis of a single leukaemia patient in terms of number of samples analysed over time, using multiple DNA and RNA sequencing methods on bulk and single cells. We provide molecular analyses of the indolent disease state, acute disease, response to therapy and remission, and finally relapse.

Because the progression of CLL is in generally slow compared with other leukaemias and solid tumours, these disease stages can be dissected and provide insights into tumour progression and evolution. While our study has the limitation of following a single patient, we believe the insights are valuable to understanding cancer.

## Methods

**Patient information.** The subject was diagnosed with CLL after a 1972 medical visit, when she presented with persistent upper respiratory infections (WBC = 9,700; 47% lymphocytes). She received antibiotics and improved, but small posterior cervical and axillary nodes, and absent splenomegaly were noted on physical exam. Eventually, she received a referral to a haematologist who performed a sternal aspiration that revealed increased lymphocytes and a picture compatible with CLL. No therapy was required. At the time of her diagnosis, two brothers and her father also had a family history of CLL. Her WBC in 1973 ranged between 15–18,000 with 80% lymphocytes, and Hgb value of 13 g dl$^{-1}$ and platelets 250–300,000 cells per μl. Physical exam at that time showed no or minimal lymphadenopathy and no splenomegaly. Two years after diagnosis a bone marrow biopsy revealed mature lymphocytes with 'focal lymphoid aggregates' consistent with at diagnosis of CLL or well-differentiated lymphocytic lymphoma. Between 2 and 6 years after diagnosis she was followed with the only physical finding noted being a very small (<1 cm) right axillary lymph node. By year 8, her WBC had risen to 35,000 cells per μl, but she remained asymptomatic without additional physical findings. By year 13, mild splenomegaly was noted, with small lymph nodes persisting (anterior and posterior cervical, and right axilla). In year 16, with WBC = 53,200 cells per μl, Hgb 11.2 g dl$^{-1}$ and platelets 114,000 cells per μl she received a short course of chlorambucil and prednisone. Her white count went from 56,000 to 25,000 cells per μl in year 17; platelets and Hgb remained stable ~100,000 cells per μl and 11.0 g dl$^{-1}$, respectively. Her spleen continued to slowly increase in size during this period and by late year 17 her counts had returned to earlier levels, WBC = 53.2 cells per μl, Hgb = 12.5 g dl$^{-1}$, platelets = 122,000 cells per μl and the spleen was measured 16 cm below the left costal margin. Otherwise, she continued to feel well with normal everyday activities. In year 19 following diagnosis she was involved in a serious car accident and suffered a splenic haematoma and other fractures (ankle and sternum). An ultrasound exam at that time revealed a 19 cm spleen. Her acute injuries and the haematoma resolved. Between year 22 and 23, she underwent another further treatment with chlorambucil and prednisone. This did not markedly change either her spleen size or WBC. By year 24, her WBC had risen to 80,000 cells per μl, Hgb 9 g dl$^{-1}$ and platelets 70,000 cells per μl. It was felt that her very large spleen was increasingly compromising her clinically, so in year 25 she underwent splenic embolization that was unsuccessful, followed by a splenectomy. The pathologic diagnosis of the spleen was 'splenomegalic CLL' and it measured (38.5 × 24.7 × 12.3 cm). She recovered and her WBC stabilized. Increasingly, she suffered from sinus and pulmonary infections, treated with rotating courses of antibiotics (levoquin, augmentin and cephlin) and monthly intravenous IG. In year 27, a breast biopsy was performed for a right breast mass and it revealed 'lymphocytes' consistent with CLL. WBC = 38,300 cells per μl, Hgb = 13 g dl$^{-1}$ and platelets = 221,000 cells per μl. In year 28, increasing pulmonary infections and bronchiectasis were noted. Lymph nodes were unchanged. WBC = 43,300 cells per μl, Hgb = 13.3 g dl$^{-1}$ and platelets = 210,000 cells per μl. The subject died in year 29, following diagnosis of increasing infectious complication of CLL. The studies were approved by the institutional review board of NIH, and informed consent was obtained from this subject.

**Cell staining for four-colour flow cytometry.** The PBMCs from the CLL patient were stained using conjugated four-colour anti-human antibodies (fluorescein isothiocyanate, FITC-CD5; phycoerythrin, PE-CD3; allophycocyanin, APC-CD19 and peridininchlorophyll-protein cyanine 5.5, PerCP-CD45) in a single tube. Briefly, the PBMC cells were thawed in a 37 °C water bath, the cells washed twice using cold Flow Cytometry Staining Buffer Solution (eBioscience), centrifuged at 3.3*g* for 1 min, re-suspended in 100 μl of staining buffer solution and transferred to 5 ml tubes (Falcon). After addition of 5 μl of each antibody to 100 μl of cells, the cells were incubated at 4 °C in the dark for 40 min, centrifuged and re-suspended in 0.5 ml of staining buffer solution and kept on ice until sorting. Controls included (1) cells only; (2) CD3 (PE); (3) CD5 (FITC); (4) CD19 (APC); and (5) CD45 (PerCP-Cy5.5). All antibodies were obtained from eBioscience.

The different target populations included (1) tumour B cells with CD19 + /CD5 + (population 7, P7); (2) normal T cells with CD3 + /CD5 + (population 5, P5); and (3) normal B cells with CD19 + /CD5 − (population 8, P8; Supplementary Fig. 2). Cells were sorted on a Becton Dickenson FACSAria II with FACSDiva v6.1 software. The cell sorter was run with a 3,050 mW 488 nm solid state laser and a 40 mW 640 nm solid state laser. The detectors for FSC, SSC had 488/10 band-pass filters. These optical filters were used in front of the fluorescent detectors: 530/30 bp (FITC), 575/25 bp (PE), 710/50 bp (PerCP-Cy5.5) and 670/30 (APC). Compensations were calculated through the auto-compensation matrix using single-stained samples. The cells were sorted using a 70-μm nozzle run at a sample flow rate of 6,000–10,000 cells per second on a sort precision of 8-32-0. Both sample and collection tubes were maintained at 4 °C during the sort. Single cells were sorted into 96-well plates with the Automated Cell Deposition Unit run at a sample flow rate of 200–400 cells per second at a sort precision of 'single cell' (0-32-16). Only the sample was maintained at 4 °C during the 96-well plate sorts. Bulk-sorted cells were viably frozen in freeze medium in a programmable cell freezer and stored in liquid nitrogen until single-cell selection was performed. In most cases, we were able to collect more than one million tumour B cells, but normal T and normal B were obtained in only small quantities.

**Sorted cell DNA extraction.** Extraction of DNA from sub-populations of cells was performed using the ZR-Duet DNA/RNA MiniPrep kit (ZYMO Research). The sample which contained normal T cells from year 10 and 28 was removed from analysis due to low quality.

**Single-cell RNA sequencing.** All complementary DNA products of single CLL B cells in this study were prepared by a microwell based a platform called MIR-ALCS[13]. A total of 1 ng complementary DNA product from each cell was used for library construction using the TruePrepTM Mini DNA Sample Prep Kit (Vazyme Biotech) followed by barcode labelling. A total of 362 single cells were sequenced by Illumina Hiseq 2000 sequencing system with SE50 sequencing strategy (Supplementary Table 4). Of these cells, 38 cells were from year 10, 48 cells from year 20, 126 cells from year 23, 35 cells from year 26 and 115 cells from year 28.

**Single-cell DNA sequencing.** Single CLL tumour cells are selected by a Micromanipulator (Eppendorf TransferMan NK2) under the inverted fluorescence microscope (Olympus IX-71), and pipetted into PCR tubes that contained 2 μl lysis buffer. Single-cell DNA amplification was carried out using the REPLI-g Single Cell Kit (Qiagen), and amplified DNA with positive results for at least six out of eight house-keeping genes were selected for subsequent library construction. DNA was fragmented with the Covaris E-210 ultrasonicator with adjusted shearing parameters. We purified the DNA fragments, blunted the ends, added A tails and ligated them with adaptors to prepare Hiseq libraries. A total of 116 single cells were sequenced on Illumina Hiseq 2000 sequencing system with PE 100 sequencing strategy (Supplementary Table 3). Of these cells, 54 cells were from year 23 and 62 cells were from year 28.

**Bulk DNA sequencing.** DNA extracted from sorted cell population of year 10, 14, 21, 23, 24, 26 and 28 by using DNeasy Blood & Tissue Kit (Qiagen). We used Covaris LE220 (Covaris) to break the genome DNA into fragments with a 350 bp peak by optimal shearing parameters. We purified the fragments by using Agencourt AMPure XP beads (Beckman Coulter). We repaired the ends of fragments, added A tails and ligated the adaptors. After fragment size selection by gel, we did 10 cycles of PCR to enrich the fragments. After purification by Agencourt AMPure XP beads (Beckman Coulter), we did the quality control of libraries by Agilent 2100 Bioanalyzer (Agilent Technologies) and quantitative PCR using ABI StepOne Plus Real-Time PCR system (Life Technologies). Paired-end 150 bp length reads were sequenced on Illumina Hiseq 4000 sequencing system.

**SNP array methods.** Genomic DNA was extracted from unsorted bulk cell DNA, as well as sorted tumour B cells, and normal T and B cells. Genomic DNA was screened and analysed at the NCI according to the standard sample handling process of the Cancer Genomics Research Laboratory, Division of Cancer Epidemiology and Genetics before genotyped using Illumina Infinium OmniExpress BeadArray assays. Sample intensity files (two files per sample, for red and green channels) were loaded into the Illumina GenomeStudio software. The intensity data were normalized using the Illumina five-step self-normalization procedure, which used information contained in the array itself to convert raw X and Y (allele A and allele B) signal intensities to normalized values. The chromosome mosaicism was detected using log R ratio (LRR) and B allele frequency (BAF). The LRR value is the normalized measure of total signal intensity and provides data on relative copy number. The BAF derived from the ratio of allelic probe intensity is the proportion of hybridized sample that carries the B allele as designated by the Illumina Infinium Assay. The LRR and BAF values for each assay were exported from GenomeStudio software using the 'Genotype Final Report' format.

Quantile normalization was applied to remove dye bias and improves the asymmetry in the detection of the two alleles for each SNP, which influences both allelic proportions and copy number estimates. GC/CPG correction was applied to reduce the wavy patterns of signal intensities and improves the accuracy of CNV detection. The LRR and BAF were re-estimated on the quantile-normalized and GC/CpG corrected values. All of these procedures were implemented using GLU software package (http://code.google.com/p/glu-genetics/) that was developed at Cancer Genomics Research Laboratory. The renormalized LRR and BAF values from qualifying assay were then analysed using custom software pipelines that involved BAF segmentation packages (http://baseplugins.thep.lu.se/wiki/se.lu.onk.BAFsegmentation) to detect mosaic copy number aberration with minimum of 20 probes per segment to minimize

the false discovery. The copy number status was assigned (mosaic loss, mosaic gain and mosaic copy-neutral LOH) and mosaic proportion of abnormal cells was estimated. All potential events were plotted. False positive calls were excluded from analysis base on manual review on each plot.

To prevent the lack of detection of the chromosome 13q14.3 focal deletion, due to the small size of this region, a t-test (two sided) was used to target the region between bases 49,139,793 and 50,269,706 (GRCh36). The LRR and BAF were examined for deviations from expected LRR and BAF. For each time point, an unequal variance t-test was applied to find mean LRR value and mean BAF for the tested region that is significantly different from mean LRR values and mean BAF on chromosome 1 (used as the reference). Results from t-test were plotted and manually reviewed to confirm a 13q14.3 focal deletion.

**Public data set access.** The human (Homo sapiens) reference genome sequence (Hg19, GRCh37) was downloaded from the University of California Santa Cruz Genome Bioinformatics (http://genome.ucsc.edu/). The transcriptome reference annotation GTF file[27] was downloaded from http://www.ensembl.org/. The GTF file retained the information of autosomes and sex chromosome X before further analysis. The known SNPs and indels in 1000 Genomes Project and dbSNP v137 (ref. 28) were downloaded from ftp://ftp.broadinstitute.org/.

**Genome alignment variant calling.** Reads resulting from Illumina Hiseq 4000 sequencing were aligned to hg19 (UCSC) using BWA (v0.7.12-r1044)[29] with parameters 'mem −t 8 − P −M' and the generated files were then sorted and PCR duplicates removed using Picard (v1.72) (http://broadinstitute.github.io/picard). Bamtools (v2.1.1)[30] was used to filter out multi-mapped reads and those with mapping quality <1. Subsequently, the BAM files were indexed by samtools (v0.1.18)[31].

**SNPs and indels calling.** We detected SNPs and indels according to the GATK[32] best practice workflows. After duplicate-read removal using Picard, the BAM files were recalibrated and variants were called using GATK v3.3.0 with known indels in the 1000 Genomes Project[33] and known SNPs in dbSNP v137 (ref. 28). Then we built the index of each VCF file, compressed them using tabix (v0.2.6)[34] and merged all SNPs in the VCF files using VCFtools (v0.1.12b)[35]. The SNPs were filtered using GATK with parameters '-T VariantFiltration --filterExpression 'QD < 2.0 || FS > 60.0 || MQ < 40.0' --filterName 'snp_filter' --clusterSize 2 --clusterWindowSize 10'.

**Minor allele frequency analysis.** Because there is no normal sample to use as a control, we could not detect the precise SNVs of the tumour samples. Therefore, we tried to eliminate possible germline SNPs and analysed the minor allele frequency changes from year 10–28. We annotated all SNPs using ANNOVAR[36] with parameters '-protocol refGene, popfreq_all_20150413, snp138NonFlagged − operation g,f,f'. The SNPs with allele frequency > 0.0001 in 1000G_AMR and ExAC_AMR data sets and the SNPs marked as snp138NonFlagged were removed from further analysis. To obtain a more confident SNP allele frequency, we retained only those SNPs with depth ≥ 30 in all seven samples. A total of 6,149 SNPs were retained. We then performed the PCA analysis using the R package FactoMineR[37], and filtered out the sites with PC1 > − 1 (Supplementary Fig. 16a,b), which showed no significant difference among each other and were thought to be possible germline SNPs.

The remained 1,065 SNPs were clustered using SOTA in R package clValid[38] and divided into 12 groups. Of the 12 clusters, 3 clusters were removed because the minor allele frequency (MAF) changed in a small range among all samples and were considered as possible germline SNPs (Supplementary Fig. 16c). Most of these SNVs are in introns and non-coding regions and are not likely to affect cancer progression. By clustering the SNVs with similar MAF pattern, we found SNVs (cluster 4) that are lost in later time points and may represent regions of deletion (Supplementary Fig. 17). Other SNVs increased in frequency and then disappeared perhaps represent variants in cells eliminated during treatment. Some SNVs showed increasing MAF during the later stages and may represent accumulated passenger mutations.

**SV detection in bulk samples.** After duplicate-read removal using Picard, structural variations were detected for each sample separately using CREST[39] with Hg19 as the reference genome. Then, the generated tumour.predSV.txt files were annotated using ANNOVAR with Hg19 and dbSNP v137. For an alternative method to call CNVs, we removed deletions and amplifications from the results of CREST. SVs with supporting reads < 3 in each sample were filtered out. But we did not detect any high confidence rearrangements of known cancer genes (Supplementary Table 8).

**Genome alignment of bulk and single-cell samples for CNV calling.** Reads resulting from Hiseq 2000 sequencing were aligned to Hg19 (UCSC) using Bowtie (v1.0.0)[40] with parameters '-S -t -m 1 --best –strata'. Then we transformed the subsequent SAM files to BAM format and sorted them using samtools

(v0.1.19). Afterwards, we also used samtools (v0.1.19) to remove PCR duplicates and build an index for each BAM file.

**CNVs calling in bulk and single-cell samples.** Copy number was computed for each sample separately using a modified method based on that developed by the Cold Spring Harbour Laboratory[41,42]. We followed their bioinformatic workflow to detect CNVs and used the Python script provided to generate the 'bin boundaries' file for 10,000 bins in Hg19 suitable for 150-bp length reads. For single-cell samples, the 'bin boundaries' file had 10,000 bins and was suitable for 100-bp length reads. We calculated the read number and the ratios based on the average read number in each bin of all bulk and single-cell samples. After GC correction, the R package DNAcopy[43] was used to calculate the copy number ratios and merge bins into segments. Bins in the same segment had the same segment ratio and segment ratios equal to 1 represent normal. The ratios > 1 represent amplification, and the rations < − 1 represent deletion. The GC corrected ratios of each bin and the inferred rations of segments of specific chromosomes were extracted to display the detailed copy number changes for bulk and single-cell samples (Supplementary Figs 9,11,12).

**Single-cell WGS filtering and clustering.** The Median Absolute Pairwise Difference (MAPD)[44] measures the absolute difference between the log2 copy number ratios of every pair of neighbouring bins and then takes the median across all bins. Higher MAPD scores reflect greater noise, typically associated with poor-quality samples. After obtaining the copy number of each bin for all single-cell-genome sequencing samples, we adopted the MAPD algorithm to filter out 14 samples with MAPD ≥ 1.4. Then, we transformed the segment ratios of the remained cells into a log2-scale, and set the log2-scaled ratios with values > 1 or < − 1 to 1 or − 1, respectively. The transformed segment ratios were used to the hierarchical clustering analysis using R package NMF[45] with the clustering method of 'ward.D2'.

**False positives in single-cell CNV profiles.** When we clustered the single cells using the CNVs, we found two cells with 6q deletions that had a larger amplification in chr12 than other cells (Fig. 3b). We extracted the raw ratios of each bin before the segment ratio calling, and found the ratios showed significant difference between the different portions in the chr12 amplification region in the same cells (Student's t-test, two sided, Supplementary Fig. 12). Although the differences of raw ratios between the true amplification region and the false positive region were significant, we speculated that the difference were not significant enough for the CNV detection methods. Therefore, the method merged the adjacent bins to the actual amplification region to generate a larger CNV, leading to a false positive result.

We also found two abnormal cells when we clustering the cells with the three main CNV regions (chr6, chr12 and chr13). We extracted the CNV profiles of the whole chromosomes, and found that both of these two cells harboured larger CNVs. One cell harboured deletion of all chr6 and 13 and is trisomy 12 (Supplementary Fig. 11a–d) and the other cell, chr6 and 13 deletion and normal chr12 (Supplementary Fig. 11e–h). We removed these two cells from the hierarchical clustering analysis of the three specific CNVs shown in Fig. 3b.

**Single-cell RNA-seq data processing.** Reads resulting from Hiseq 2000 sequencing were filtered using an in-house C + + script, and aligned using TotHat2 (ref. 46). The fragments per kilobase of exon per million fragments mapped (FPKM) values were calculated using the R package edgeR[47]. The detailed process and parameters were described in a recent publication[13]. The single cells with less than one million reads mapping to gene regions were removed from further analysis.

**Filtering of genes and samples.** Cells with expressed genes (FPKM > 1) > 8,000 were also filtered out, as they showed much higher expression levels than the average and likely contain more than one cell. To reduce the potential influence of RNA degradation, we only retained the protein-coding genes. We then picked a set of genes that were expressed in at least half of all cells with FPKM > 1 and calculated the expressed gene number of every cell. We excluded from the analysis any cells that expressed below the 10th percentile of that gene set and whose mean FPKM ≤ 1. We also removed genes with the mean FPKM more than the 99th percentile for all genes. In total, 300 single cells and 9,258 genes remained for further analysis out of 362 total cells sequenced.

**PCA and hierarchical clustering analysis.** We used an in-house R script to perform the PCA and hierarchical clustering. The PCA analysis was performed on all the cells and genes after filtering using the R package FactoMineR[37]. A total of 80 genes, the first 20 genes of the first four principle components, were used to perform the hierarchical clustering. The cells were divided into six groups according to the result of the hierarchical clustering. We then performed the Kruskal–Wallis rank sum test (KW test) and adjusted the P values using 'fdr' method. We retained the genes with q value < 0.01. The remaining genes were used to perform the PCA, hierarchical clustering analysis and KW test. Genes with q value < 0.01 were used for the third PCA and hierarchical clustering analysis, which

were the final results used in the paper (Fig. 4a). The cells were then divided into six groups (A–F cluster) according to the hierarchical clustering, and we used the hypergeometric distribution test to identify the marker genes. The *P* values were adjusted using the 'BH' method, and genes with *q* values < 0.01 were considered as particularly highly expressed marker genes in the A–F clusters (Supplementary Table 5). We did not detect significantly expressed genes in cluster C.

**Pseudo-temporal analysis.** To further investigate the gene expression changes during tumour evolution and the relationships between different cells from different time points, we performed pseudo-temporal analysis using the R package Monocle[19]. We first detected the differentially expressed genes according to corresponding time point of each cell using Monocle and 3,546 genes with a *P* < 0.01 were retained for further analysis. The pseudo-temporal path number was set at 2, and the cells were divided into three trajectories. Considering that the cells were not in a strict developmental state and that the intervals between different time points were very long, we modified the pseudo-temporal ordering of each cell. Cells were divided in to three groups according to the inferred pseudo-temporal phases, and the relative orderings of cells in the same group remained unchanged. The group containing the most cells from the earliest time (year 10) was set to be the start of the pseudo-temporal path, and the group containing the most cells from year 23 and 28 was set as the end of the path (Fig. 4c,d). The last group was placed between the two groups. We then clustered the genes into 10 groups to make sure that the gene expression along the inferred pseudo-temporal path reflected the changes along the actual time (Supplementary Fig. 15). Therefore, we could combine the pseudo-temporal ordering and the actual time to investigate the gene expression changes and their potential influences. Reactome enrichment was performed on the genes from each group using the R package ReactomePA[48] with the 9,258 genes as the background and a *q* value cutoff of 0.05 (Supplementary Table 7). We did not detect enriched pathways in cluster 6.

**Data availability.** Raw data have been uploaded to the database of Genotypes and Phenotypes (dbGAP, http://www.ncbi.nlm.nih.gov/gap/) under the study accession code phs001177, SRA399097. This study includes all sequencing data from WGS and whole-transcriptome sequencing. The remaining data supporting the finding of this study are contained within the article and the Supplementary Information files, or available from the authors on request.

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

## Acknowledgements

We thank Rongchang Chen, Weijian Rao, Xiaolong Zhang, Jie Wang, Zhanlong Mei, Xinlan Zhou, Nannan Li, Xulian Shi and Fatima Abbassi for help on data analysis and discussion, Kathleen Noer and the CCR Flow Cytometry Core for cell sorting. This project was supported by grants of the Shenzhen Science and Technology Program (CXZZ20150330171838997), the Shenzhen Municipal Government of China (ZDSYS20140509153457495) and the Science, Technology and Innovation Committee of Shenzhen Municipality (JSGG20140702161347218). This project has been funded in whole or in part with federal funds from the National Cancer Institute, National Institutes of Health, under Contract No. HHSN261200800001E. The content of this publication does not necessarily reflect the views or policies of the Department of Health and Human Services, nor does mention of trade names, commercial products or organizations imply endorsement by the US Government.

## Author contributions

Y.H., N.C. and M.D. conceived this project. S.L., B.L., M.D. and Y.H. supervised this project. M.D. and N.C. provided the samples. Z.Z., L.G., W.Z., S.L., Q.Y. and M.J. analysed the data. G.M. was one of the physicians caring for this patient and his laboratory performed several of the flow cytometric analyses. Z.Z., M.J., L.G., W.Z., Q.Y. and S.L. draw the figures. Q.Y., Z.Z., M.J. and F.L. performed the variation analyzes. L.W., Y.L., X.L., Z.W., G.L., K.W. and L.X. performed the sample amplification and library construction. Z.Z., L.G., S.L, Q.Y., Y.H. and M.D. wrote the manuscript with the input from the remaining authors. B.L., G.M., H.L., S.T., M.M., M.Y., K.W., H.Y., X.X., S.C., N.C. and M.D. reviewed the manuscript.

## Additional information

**Competing financial interests:** The authors declare no competing financial interests.

