## [Peer Review File · Nature Communications]

REVIEWERS' COMMENTS:

Reviewer #1 (Remarks to the Author):

The authors have addressed all my previous comments and criticisms. Needless to say, the study has the obvious limitation of following a single CLL patient, but some findings may be of general interest.

Reviewer #2 (Remarks to the Author):

In this paper the genetic evolution of the disease was followed in a patient diagnosed with CLL over a 29-year time period. By applying genomic arrays on multiple time point samples they could depict major genetic events/changes occurring at the different phases of the disease. Whole-genome sequencing (WGS) and single cell WGS/RNA-seq were also performed for an in-depth analysis of clonal dynamics.

Major comments:

Despite the fact that they have analysed an impressive number of samples from this patient and applied both established and novel high-resolution techniques, my major concern is the lack of novelty for CLL patients in general.

Detailed longitudinal analysis has been published in considerably higher number of patients (though admittedly fewer samples) and there is no obvious new genomic data from this paper that further our understanding significantly compared to these studies.

This patient had a very indolent disease course, survived for almost 3 decades and only received an old type of treatment (chlorambucil). It is hence unclear what we can learn from this extreme patient for CLL patients at large.

The only novel part is the single-cell analysis (RNA-seq/WGS) at selected time points. This reviewer would advise them to focus on this aspect and even consider to publish the SNP-array data separately.

Since they have performed WGS it would be very valuable to see how the somatic mutation patterns changed over time. They explain in the Methods section that no normal sample was available to allow this, however why did they not use the remission sample obtained year 27 as control?

Minor comment:

In the introduction it is mentioned that single-cell WES has been performed but no data is provided.

CLL paper NC point by point

Reviewer #1

The authors have addressed all my previous comments and criticisms. Needless to say, the study has the obvious limitation of following a single CLL patient, but some findings may be of general interest.

We thank the reviewer for the careful consideration and comments.

Reviewer #2

In this paper the genetic evolution of the disease was followed in a patient diagnosed with CLL over a 29-year time period. By applying genomic arrays on multiple time point samples they could depict major genetic events/changes occurring at the different phases of the disease. Whole-genome sequencing (WGS) and single cell WGS/RNA-seq were also performed for an in-depth analysis of clonal dynamics.

Major comments:

Despite the fact that they have analysed an impressive number of samples from this patient and applied both established and novel high-resolution techniques, my major concern is the lack of novelty for CLL patients in general.

Detailed longitudinal analysis has been published in considerably higher number of patients (though admittedly fewer samples) and there is no obvious new genomic data from this paper that further our understanding significantly compared to these studies.

This patient had a very indolent disease course, survived for almost 3 decades and only received an old type of treatment (chlorambucil). It is hence unclear what we can learn from this extreme patient for CLL patients at large.

The only novel part is the single-cell analysis (RNA-seq/WGS) at selected time points. This reviewer would advise them to focus on this aspect and even consider to publish the SNP-array data separately.

Since they have performed WGS it would be very valuable to see how the somatic mutation patterns changed over time. They explain in the Methods section that no normal sample was available to allow this, however why did they not use the remission sample obtained year 27 as control?

We thank the reviewer for the advice. While this is a single patient and longitudinal studies have CLL have been published, no other study (for CLL or other cancers) provides as comprehensive an analysis as we present.

Except for selected timepoints, the WGS was low coverage and not sufficient to accurately call mutations. This particular patient has a very limited number of somatic mutations. In our opinion, the year 27 sample would not be an effective control, but we feel our methods to detect somatic mutations are not hampered by a lack of normal cells.

The second to last paragraph of the Discussion now outlines many of the limitations of the study including that it involves a single patient with a mostly indolent course of disease.

Minor comment:

In the introduction it is mentioned that single-cell WES has been performed but no data is provided.

This incorrect reference has been removed.